# Influence of Salting on Physicochemical and Sensory Parameters of Blue-Veined Cheeses

Noemí López González, Daniel Abarquero [ID], Patricia Combarros-Fuertes [ID], Bernardo Prieto, José María Fresno [ID] and María Eugenia Tornadijo *

Department of Food Hygiene and Technology, Faculty of Veterinary Science, University of Leon, 24071 León, Spain; nlopeg01@estudiantes.unileon.es (N.L.G.); dabac@unileon.es (D.A.); pcomf@unileon.es (P.C.-F.); bprig@unileon.es (B.P.); jmfreb@unileon.es (J.M.F.)
* Correspondence: metorr@unileon.es; Tel.: +34-987-293-253

**Abstract:** Salting influences microbial growth, enzymatic activity, and biochemical reactions during ripening, thus contributing to the final quality of cheese. The aim of this study was to evaluate the influence of different salting methods (dry salting at 12, 24, and 48 h; salting in brine; and salting of partially drained curd, before moulding) on the chemical (moisture and salt content), physicochemical (pH, titratable acidity, and water activity), and sensory characteristics (texture profile analysis, colour, and sensory attributes) of industrial blue-veined cheese. Dry-salted cheeses had lower moisture content and water activity, and higher salt/moisture content and acidity than those salted in brine and in the partially drained curd. Dry-salted cheeses were also characterised by higher values for hardness, fracturability, and instrumental gumminess. Dry-salted cheeses showed differences only in the red/green colour component (a*), with the cheeses salted in the partially drained curd being less greenish. All cheeses scored high (around 7) in the tasters' overall impression, with the dry-salted cheeses at 12 and 24 h showing optimal growth and distribution of mould, as well as better flavour and texture.

**Keywords:** blue-veined cheese; salting; salt/moisture ratio; sensorial; texture; water activity





## 1. Introduction

Salting of cheese can be carried out by various methods that affect the absorption and diffusion of salt into the cheese [1]. (i) Salting of cheese in brine involves the movement of NaCl molecules ($Na^+$ and $Cl^-$ ions) from the brine into the cheese because of the difference in osmotic pressure. Water diffuses out of the cheese to restore the osmotic pressure balance. NaCl, like water, moves in response to its concentration gradient, as it would in pure solutions, but its diffusion rate is slower. (ii) Another method is to spread the salt over the surface of the partially drained curd, before moulding, in which case some of the NaCl dissolves in the surface moisture and slowly diffuses into the interior. This, in turn, causes whey from the curd to diffuse to the surface, dissolving the salt crystals and creating a supersaturated salt solution around each particle, accelerating salting, although some of the brine formed on the surface of the curd particles is lost during subsequent moulding and pressing of the cheese. (ii) In the method of salting on the surface of the cheese, it is necessary to deposit a layer of salt on the surface of the cheese which forms with the whey a layer of concentrated brine. There is considerable shrinkage of the cheese surface with significant loss of surface moisture and reduced mobility of NaCl into the interior, resulting in a lower salt incorporation rate compared with salting in brine. Regardless of the method used, the salting of cheese results in the diffusion of salt into the cheese and the outflow of water by osmosis. This facilitates syneresis, which helps to remove water from the cheese. Salting also enhances the flavour of the cheese and helps to reduce its water activity (aw). The reduction in aw affects the biochemical processes that take place during ripening [2].

Blue cheeses are usually dry-salted by rubbing, which consists of spreading coarse salt (in a dose of about 4%) on the surface of the cheese, either manually or mechanically, twice over a period of 5 days. This operation is carried out in cold conditions (around 10–12 °C) after the cheese has been moulded and the curd has been drained for 2 to 4 days at 18–20 °C. However, some cheesemakers add salt to the partially drained curd before the moulding to facilitate distribution or dip the cheeses in brine, in the latter case at temperatures of around 10 °C [3], to standardize the process and reduce labour. In some cases, salting is also carried out by adding salt to the milk [4]. The salt content of blue cheese is usually between 3 and 4% of the total weight of the cheese [5]. Values in the range of 1.5 to 5% of the total weight of the cheese have also been described. In fact, some values of salt content in blue cheeses that have been described refer to Picón Bejes-Tresviso, with 1.9% NaCl [6], Gamonedo with 4.9% [7], or Valdeón cheese aged 4 months with 5.33% [8].

During the first stages of the ripening of blue cheeses, aw is high, around 0.990, which allows the growth and activity of starter cultures. At the end of the ripening process, blue cheeses are characterised by aw values between 0.880 and 0.925 [6,9–11]. As mentioned above, NaCl contributes to the decrease in cheese moisture and to the reduction in aw, a decrease to which low molecular weight compounds (peptides and amino acids) generated by proteolysis during the ripening process also contribute. Salt concentration, together with other factors such as aw, pH, and temperature, play an important role in the growth and development of starter cultures, both lactic acid bacteria (LAB) and thus in acidification, as well as in the development of *Penicillium roqueforti* during blue cheese ripening [12]. *P. roqueforti* can grow in a range of aw between 0.998 and 0.840, with its adaptation phase being relatively stable at aw values above 0.920 but increasing dramatically at values below 0.920. Therefore, the final aw values should allow *P. roqueforti* to grow as quickly as possible and to grow during all stages of cheese making, including ripening. On the other hand, salt content, together with pH and calcium concentration, influences the water-holding capacity of the protein matrix and its tendency to syneresis. It also influences texture, as an increase in salt content leads to an increase in hardness and a decrease in cohesiveness due to a lower degree of proteolysis and hydration of casein [13]. In addition, salting can control the development of bitter flavours in cheeses in relation to the inhibitory effect on β-casein proteolysis exerted by concentrations of 5% NaCl or higher [13]. In fact, studies experimenting with alternative methods of replacing NaCl with KCl have reported an increase in bitterness and difficulty in properly ripening the cheese [14]. Finally, it should be noted that salting influences cheese yield. The amount of whey released is directly related to the amount of salt added. This leads to a significant reduction in the weight of the cheese, which can be as much as 3%, making it essential to carry out the salting process in a standardised and controlled manner to avoid economic losses during the production process. In view of the above, the aim of this study was to investigate the influence of the salting process (dry salting at 12, 24, and 48 h, salting in brine, and salting in the partially drained curd, before moulding) on some chemical, physicochemical, and sensory parameters of a blue-veined cheese after one month of ripening, and to identify the most advantageous production process that produces a positive impact on the quality of this cheese.

## 2. Materials and Methods

### 2.1. Cheese Production

The cheeses used in this study were produced at the Quesos La Peral, S.L. cheese factory (La Peral-Illas, Asturias, Spain), according to the usual production process described below. The starting point was 6000 L of cow's milk, the basic chemical composition of which was determined using Milkoscan equipment (Foss, Denmark). The composition was 12.59% dry matter, 3.84% fat, 3.26% protein, and 4.64% lactose.

The pasteurised milk was kept in the vat at a temperature of 32 °C, pH 6.58. Then, 1 L of 30% (*w/v*) calcium chloride solution (Biostar, Spain), 6 packages (500 U each) of freeze-dried starter cultures of CHN19 and Flora Danica (CHR Hansen, France), consisting

of mesophilic LAB strains (*Lactococcus lactis*, *L. cremoris*, *L. lactis* subsp. *diacetylactis*, and *Leuconostoc* spp.), and finally 20 mL of a PRB6 suspension of *Penicillium roqueforti* (Danisco, France). The milk was kept at 32 °C with moderate shaking until a pH of 6.50 was reached. The milk was then coagulated by adding 1 L of calf extract rennet (180 IMCU/mL; chymosin/pepsin: 80/20%; Biostar, Austria). After 1–1.5 h and with a pH of 6.42, the curd was cut to a grain size of approximately 1.5 × 1.5 cm and then gently stirred for 10 min. Then, in the first stage, 600 L of whey was withdrawn, and the curd was stirred for about 15 min, followed by a second stage in which a further 600 L of whey was withdrawn, and the curd was stirred for a further 15 min until a pH of 6.25 was reached. At this point, 4 kg of partially drained curd was removed from the vat to which 2% (*w/w*) of salt was added; the salt was mixed with the curd to facilitate its homogeneous distribution, at a temperature of 32 °C. The salted curd was then placed in two cylindrical moulds without a base, 18 cm high and 13 cm in diameter, to produce two cheeses weighing approximately 800 g, 8 cm high, and 12 cm in diameter, which made up the batch of partly drained and salted curd prior to moulding (batch L5). The rest of the unsalted curd remaining in the vat was placed on a sieve and then manually placed in cylindrical moulds without a base, 18 cm high and 13 cm in diameter. The moulds with the unsalted curd were transferred to the draining chamber at a temperature of 20 °C and 83% relative humidity, where they remained for a certain time, depending on the batch, and were turned three times a day.

Eight cheeses of approximately 800 g, 8 cm high, and 12 cm in diameter were subjected to different salting methods: dry salting for 12 h (batch L1); dry salting for 24 h (batch L2); dry salting for 48 h (batch L3), and salting in brine (18° Baumé solution at a temperature of 8–10 °C) (batch L4). For the 12 h dry-salted cheeses, the first salting (one side of the cheese and its contour) was carried out 12 h after moulding, followed the next day by the second salting (the other side of the cheese and its contour). For the 24 h dry-salted batch, the first salting was carried out 24 h after moulding, followed by the second salting the next day. For the 48 h dry-salted batch, the first salting was carried out 48 h after moulding and the second salting was carried out the following day. The salted cheeses in brine were placed in brine after 36 h of moulding and remained in brine for a further 36 h. All batches of cheeses were then transferred to a chamber at 8 °C and 90% relative humidity where they were pricked after 5 days. Then, cheeses were transferred to the ripening chamber where they remained for 28 days at a temperature of 10 °C and a relative humidity of 84%. The process described above is shown in Figure 1. Finally, the cheeses were wrapped in food-grade aluminium foil and transported under refrigeration (4 °C) to the laboratory for analysis. At the laboratory, the cheeses were weighed and one-third was peeled, crushed, and vacuum-packed in 30 g bags, while the rest was left intact. In both cases, the samples were kept at 4 °C until analysis, which was carried out in all cases within one week of collection.

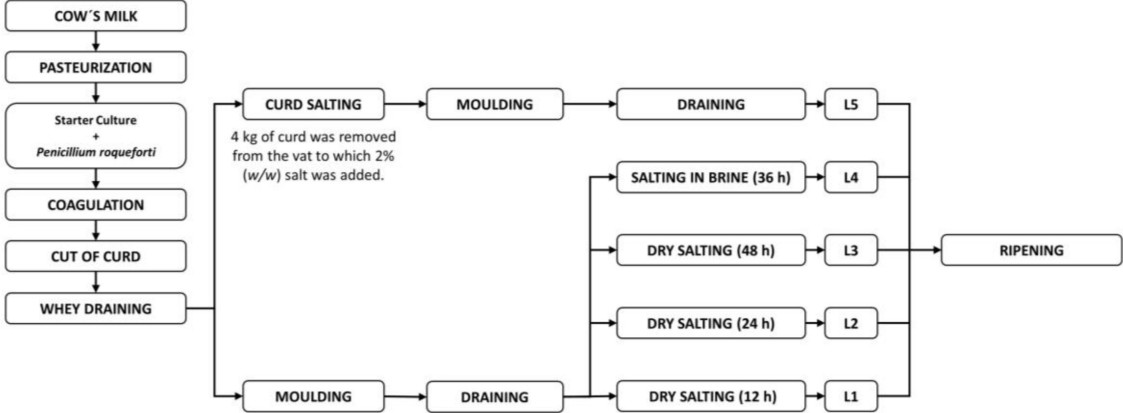

**Figure 1.** The production process and the methods of salting used for the five batches of blue-veined cheeses.

### 2.2. Chemical and Physicochemical Analysis

The moisture content of the cheese was determined by drying in a hot forced-air oven to constant weight according to ISO 5534 [15] and expressed as a percentage by weight of the sample. The determination of sodium chloride in cheese was carried out according to the method described by Vohlard in accordance with AOAC standard 935.43 [16] and expressed as g NaCl in 100 g of cheese. The salt/moisture ratio was determined based on the NaCl content, expressed as g of salt per 100 g of moisture. The pH of the cheese was determined according to AOAC 14.022 [17], while the titratable acidity was determined according to AOAC 162.47 [18] and expressed as the percentage of lactic acid on the dry cheese extract. Water activity (aw) was determined using the AQUA LAB dew point analyser CX-2 (Decagon Devices Inc., Pullman, WA, USA).

### 2.3. Measurement of the Colour of Cheeses

Instrumental colour measurement was performed using a Konica CM-700d reflectance colorimeter (Minolta, Osaka, Japan) according to the method described by Diezhandino et al. [19]. A MAV reading using a MAV standard mask with a diameter of 8 mm with glass was carried out. The data processing software was CM-S100w SpectraMagic TM NX version 1.9, Pro USB (Konica, Minolta, Osaka, Japan). For colour analysis, approximately 1 cm thick longitudinal samples of cheese were taken and a total of 12 measurements were made at different points in each sample. For all colour determinations, the instrument was previously calibrated for zero and white using the standard plates supplied by the manufacturer, the D65 illuminant, and the $10°$ SCI observer. The colour parameters of the CIELab scale studied were luminance (L*), red–green component (a*), and yellow–blue component (b*).

### 2.4. Texture Profile Analysis

The instrumental texture parameters, including fracturability (N), hardness (N), adhesiveness (N·s), cohesiveness, springiness, gumminess, and chewiness, were determined using a Stable Micro Systems Texturometer mod. TZ-XT2i (Aname, Madrid, Spain), using the instrumental analysis of the texture profile described by Bourne [20]. Nine replicates (15 mm in diameter and 19 mm in height) were obtained for each cheese using a die from which 5 mm of rind had previously been removed. The analysis was carried out at a temperature of $20 \pm 2$ °C using a plate–plate system with an SMS P/75 stainless steel probe at a constant speed of 0.5 mm s$^{-1}$ and compression of the samples to 80% by two compression cycles. The results were analysed using Texture Expert Stable Micro Systems software, version 2.64 (Aname, Madrid, Spain).

### 2.5. Sensory Analysis

The sensory analysis of the cheeses was carried out using ranking tests in accordance with the ISO 8587 standard [21] to determine whether the different salting methods used influenced the following sensory characteristics of the cheeses: degree of mouldiness, homogeneity of the blue vein, odour, flavour, degree of saltiness, and hardness. For this purpose, a panel of 31 tasters (most of them with experience in cheese tasting), was asked to rank the cheese samples from each batch in a hierarchical order (1–5), from most preferred to least preferred (the value of 1 corresponding to the first preferred cheese batch and the value of 5 corresponding to the last preferred cheese batch). Finally, the same tasters carried out an overall impression test to measure the degree of influence of the type of salting on the acceptability of the cheeses, using a scale from 1 (very bad) to 10 (excellent).

### 2.6. Statistical Analysis

Statistical analysis was performed using SPSS v.21 (SPSS, Chicago, IL, USA). The variables studied were checked for normality and homoscedasticity. A one-way ANOVA was then performed using the Tukey HSD test at a 5% significance level ($p < 0.05$). In the case of sensory analysis, Friedman's test was used to treat the order and overall impression test data.

## 3. Results

*Chemical and Physicochemical Parameters*

Table 1 shows the values of chemical and physicochemical parameters obtained for the different cheese batches. In general, the results obtained showed significant differences ($p < 0.05$) in the moisture, salt, salt/moisture, and aw contents depending on the salting process used in the production of the cheeses, which a priori seems to demonstrate the influence of this stage in the cheese ripening process and, therefore, on the characteristics of the final product.

**Table 1.** Chemical and physicochemical parameters (mean ± standard deviation of the data) of the blue-vein cheeses after one month of ripening time.

| | L1 | L2 | L3 | L4 | L5 |
|---|---|---|---|---|---|
| Moisture (g 100 g$^{-1}$ cheese) | 41.42 [a] ± 0.87 | 40.07 [b] ± 0.14 | 39.72 [b] ± 0.30 | 45.16 [c] ± 0.33 | 45.60 [c] ± 0.79 |
| Salt content (g 100 g$^{-1}$ cheese) | 1.86 [a] ± 0.02 | 1.84 [a] ± 0.022 | 2.15 [b] ± 0.01 | 1.97 [c] ± 0.01 | 1.15 [d] ± 0.21 |
| Salt/moisture (g 100 g$^{-1}$ moisture) | 4.50 [ac] ± 0.12 | 4.59 [a] ± 0.04 | 5.42 [b] ± 0.04 | 4.35 [c] ± 0.04 | 2.51 [d] ± 0.04 |
| aw | 0.972 [a] ± 0.001 | 0.974 [a] ± 0.003 | 0.971 [a] ± 0.001 | 0.982 [b] ± 0.002 | 0.992 [c] ± 0.002 |
| Titratable acidity (g 100 g$^{-1}$ total solids) | 2.36 [a] ± 0.07 | 2.26 [a] ± 0.10 | 2.27 [a] ± 0.20 | 1.42 [b] ± 0.12 | 1.71 [c] ± 0.07 |
| pH | 5.76 [a] ± 0.08 | 5.87 [b] ± 0.02 | 5.96 [c] ± 0.03 | 6.15 [d] ± 0.05 | 5.77 [acd] ± 0.23 |

L1: batch dry-salted at 12 h; L2: batch dry-salted at 24 h; L3: batch dry-salted at 48 h; L4: batch salted in brine; and L5: batch salted in partially drained curd. Mean ± standard deviation of the data obtained from the analysis of 3 replicates per cheese and two cheeses per batch ($n$ = 24). Values with different superscripts were significantly different ($p < 0.05$) between batches for each parameter studied.

The highest moisture content was obtained in cheeses salted in partially drained curd (L5), followed by those salted in brine (L4) and the dry-salted cheeses (L1–L3). Moisture was very similar between cheeses salted for 24 h (L2) and 48 h (L3) but significantly different ($p < 0.05$) from cheeses salted for 12 h (L1). In general, it could be observed that the higher the salt content of the cheeses, the lower the moisture values, except in the case of cheeses salted in brine (L4). As shown in Table 1, the moisture content of the cheeses salted in brine was about 9–12% higher than that of the dry-salted cheeses, although it was similar ($p > 0.05$) to that of the cheeses salted in curd (L5).

Table 2 shows the colour parameters for the different batches of blue cheese. In general, it could be found that the type of salting applied to the cheeses did not have a significant influence ($p > 0.05$) on the CIELab parameters of the blue cheeses, except for the a* parameter. The luminosity (L*) values were very similar, ranging from 83.08 for cheeses salted dry for 48 h (L3) to 86.30 for cheeses salted in brine (L4). A very similar behaviour was also observed for the values obtained for the yellow/blue coordinates (b*), with all blue cheeses showing values between 12.6 and 13.9. The only significant differences ($p < 0.05$) were described for the red/green coordinates (a*), with dry-salted blue cheeses presenting negative values between −0.5 and −1.05, which was different from the contents in the salted samples in brine (L4) (+0.51); while the cheeses that had been salted in the partially drained curd (L5) presented values around 0.

**Table 2.** Colour (mean ± standard deviation of the data) of the blue vein cheeses after one month of ripening time.

| | L1 | L2 | L3 | L4 | L5 |
|---|---|---|---|---|---|
| L* | 84.52 ± 3.20 | 83.49 ± 3.38 | 83.08 ± 3.90 | 86.30 ± 2.96 | 85.69 ± 3.64 |
| a* | −0.94 [a] ± 1.12 | −0.54 [a] ± 0.98 | −1.08 [a] ± 1.06 | 0.51 [b] ± 0.68 | −0.03 [ab] ± 0.81 |
| b* | 13.66 ± 1.52 | 13.78 ± 1.90 | 13.27 ± 1.21 | 13.95 ± 1.25 | 12.61 ± 1.47 |

L1: batch dry-salted at 12 h; L2: batch dry-salted at 24 h; L3: batch dry-salted at 48 h; L4: batch salted in brine; and L5: batch salted in partially drained curd. Values with different superscripts were significantly different ($p < 0.05$) between batches for each parameter studied. Mean ± standard deviation of the data obtained from the analysis of 12 replicates per cheese and two cheeses per batch ($n$ = 24).

Table 3 shows the results obtained for the main parameters studied in the analysis of the texture profile of the different blue cheeses. It was observed that the salted blue cheeses in brine (L4) and in partially drained curd (L5) had lower values for friability, hardness, gumminess, and chewiness compared with the salted dry cheeses (L1, L2, and L3). However, no differences were found in the values for cohesiveness and adhesiveness. The lowest values obtained for gumminess, chewiness, and hardness were presented by the batches L5 and L4, which is related to their moisture content. In the case of adhesiveness, which measures the force required to separate the cheese sample from the probe, no significant differences ($p > 0.05$) were observed in the values obtained for the different batches of blue cheeses analysed. Finally, it should be noted that no significant differences ($p > 0.05$) were found in the cohesiveness values, whereas they were found, albeit minimally, in the springiness values.

**Table 3.** Texture parameters (mean $\pm$ standard deviation of the data) obtained in the analysis of the blue cheeses after one month of ripening time.

|  | L1 | L2 | L3 | L4 | L5 |
|---|---|---|---|---|---|
| Fracturability (N) | 55.23 [a] $\pm$ 3.25 | 47.10 [b] $\pm$ 4.23 | 43.54 [b] $\pm$ 2.92 | 25.32 [c] $\pm$ 3.07 | 27.02 [d] $\pm$ 1.26 |
| Hardness (N) | 167.63 [a] $\pm$ 9.44 | 150.22 [ab] $\pm$ 8.63 | 134.55 [b] $\pm$ 9.97 | 109.66 [c] $\pm$ 6.95 | 72.9 [d] $\pm$ 10.82 |
| Adhesiveness (N·s) | $-4.29 \pm 1.18$ | $-3.49 \pm 0.83$ | $-3.78 \pm 0.87$ | $-3.93 \pm 1.36$ | $-5.28 \pm 1.44$ |
| Cohesiveness | $0.11 \pm 0.01$ | $0.11 \pm 0.00$ | $0.11 \pm 0.01$ | $0.11 \pm 0.1$ | $0.11 \pm 0.01$ |
| Springiness (%) | 0.25 [a] $\pm$ 0.04 | 0.18 [b] $\pm$ 0.03 | 0.15 [b] $\pm$ 0.01 | 0.19 [bc] $\pm$ 0.03 | 0.22 [c] $\pm$ 0.02 |
| Gummyness (N) | 18.66 [a] $\pm$ 0.90 | 16.53 [ab] $\pm$ 1.14 | 15.13 [b] $\pm$ 1.41 | 11.78 [c] $\pm$ 0.98 | 7.74 [c] $\pm$ 1.92 |
| Chewyness (N) | 4.59 [a] $\pm$ 0.65 | 2.99 [b] $\pm$ 0.67 | 2.28 [bc] $\pm$ 0.36 | 2.21 [bc] $\pm$ 0.48 | 1.72 [c] $\pm$ 0.61 |

L1: batch dry-salted at 12 h; L2: batch dry-salted at 24 h; L3: batch dry-salted at 48 h; L4: batch salted in brine; and L5: batch salted in partially drained curd. Values with different superscripts were significantly different ($p < 0.05$) between batches for each parameter studied. Mean $\pm$ standard deviation of the data obtained from the analysis of 9 replicates per cheese and two cheeses per batch ($n = 18$).

Table 4 shows the cumulative scores obtained in the grading test for the degree of mouldiness, homogeneity of the blue vein, odour, flavour, degree of saltiness, and hardness. The overall impression, mean scores, and standard deviations obtained after sensory analysis by a panel of 31 tasters aged between 18 and 63 are also shown. In the case of the ordering test, lower values indicate a greater preference on the part of the tasters for the sensory attribute evaluated.

**Table 4.** Results of the sensory analysis of blue vein cheeses ripened for one month using an ordering test.

|  | L1 | L2 | L3 | L4 | L5 |
|---|---|---|---|---|---|
| Degree of mould development | 71 [a] | 66 [a] | 116 [b] | 106 [b] | 106 [b] |
| Homogeneity of blue veins | 61 [a] | 68 [a] | 108 [b] | 118 [b] | 110 [b] |
| Odour | 92 | 93 | 95 | 93 | 92 |
| Flavour | 70 [a] | 97 [b] | 105 [b] | 101 [b] | 90 [ab] |
| Degree of saltiness | 74 [a] | 98 [ab] | 104 [b] | 99 [b] | 87 [ab] |
| Hardness | 67 [a] | 77 [a] | 114 [b] | 116 [b] | 88 [a] |
| Overall impression | $6.50 \pm 1.72$ | $7.00 \pm 1.74$ | $7.30 \pm 1.72$ | $6.80 \pm 1.45$ | $6.75 \pm 1.99$ |

L1: batch dry-salted at 12 h; L2: batch dry-salted at 24 h; L3: batch dry-salted at 48 h; L4: batch salted in brine; and L5: batch salted in partially drained curd. Values with different superscripts were significantly different ($p < 0.05$) between batches for each parameter studied. Ordering test: The scores correspond to the sum of the scores given by the tasters for each parameter. Lower values indicate a greater preference on the part of the tasters for the sensory attribute evaluated.

The degree of development of blue mould and the homogeneity of the blue vein in the cheese mass showed significant differences ($p < 0.05$) between the different batches, with the best values corresponding to the cheeses dry-salted at 12 h (L1) and 24 h (L2). On the other hand, cheeses salted in partially drained curd (L5), in brine (L4), and dry-salted after

48 h (L3) showed similar values for both attributes, with no significant differences ($p > 0.05$) between the tasters' samples. Odour was the only attribute that did not show significant differences between batches ($p > 0.05$), as it was rated very similarly by the tasters for all the cheeses studied. However, they did show significant differences ($p < 0.05$) in flavour. The tasters preferred the cheeses salted dry after 12 h (L1) and those salted in the curd (L5), although the latter did not show significant differences ($p > 0.05$) with respect to the other cheeses studied (L2, L3, and L4). Finally, the overall impression was very similar for all the cheeses, ranging from 6.5 to 7.3 (out of 10), with no significant differences between cheeses ($p < 0.05$).

## 4. Discussion

The cheeses obtained by being salted in partially drained curd (L5) had between 39 and 47% less salt than the dry- and brine-salted cheeses. According to the experimental model of Breene et al. [22], fat exuded on the surface of the curd grains prevents the absorption of salt. Consequently, the salt remains largely dissolved in the whey and is lost during the subsequent whey draining stage. In the case of the batch salted in brine (L4), the low temperature (around 10 °C) of the brine could have limited the diffusion of salt into the cheese, as the fat would be below its melting point and would solidify on the surface of the granules. However, this effect could be counteracted by a lower resistance to the diffusion of salt molecules in the cheese matrix (less dense and more porous) [23]. On the other hand, as shown in Table 1, the moisture contents of the different dry-salted cheeses (L1, L2, and L3) were lower than those of those salted in brine (L4), although the salt content was similar in all of them, with no significant differences ($p > 0.05$) in the dry-salted cheeses at 12 (L1) and 24 h (L2). One explanation could be that because of the osmotic gradient created during dry salting, there was a greater flow of water to the outside of the cheeses, which hindered the diffusion of salt into the cheese. Other authors reported that dry-salted cheeses had higher moisture content, lower salt content, and a lower salt/moisture ratio than those salted in brine, which was attributed to a greater loss of salt during the salting process [13,24].

The differences observed in the salt concentration between the cheeses salted for 48 h (L3) and those dry-salted for 12 h (L1) and 24 h (L2) could be related to the degree of draining of the curds before salting. The curds corresponding to the cheeses salted for 12 h (L1) and 24 h (L2) had shorter draining times in the moulds and pH values slightly higher than the isoelectric point of the caseins compared with those salted for 48 h. These 48 h curds had an open structure and pH values of around 4.6–4.8, which contributed to increasing the electrostatic interactions between the caseins and therefore removing more water during draining. In fact, Fox et al. [2] reported that the intensity of the syneresis of the curds during the draining process was directly related to the degree of acidity and inversely related to the pH. Consequently, the water flow established in the dry-salted cheeses at 48 h (L3) resulted in the formation of concentrated brine on the cheese surface, facilitating the outflow of whey and further diffusion of salt into the cheese. The higher shrinkage of the cheese in the area close to the salt, because of the effect of the salt on the proteins, leads to a higher loss of moisture from the cheese. This effect limits the diffusion of the salt into the cheese [25].

The salt/moisture ratio is a fundamental parameter for interpreting the final quality of ripened cheeses due to its influence on the development of the microbiota as well as on the enzymatic activity, which determines the final acidity, pH, and aw values of the cheeses. Both the dry-salted cheeses after 12 h (L1) and 24 h (L2) and those salted in brine (L4) presented very similar salt/moisture values (around 4.35–4.59), although all of them showed significant differences ($p < 0.05$) with those salted in partially drained curd (L5) and dry-salted cheeses after 48 h (L3). Curd-salted cheeses (L5) showed the lowest salt/moisture values (2.51%), while dry-salted cheeses after 48 h (L3) showed the highest values (5.42%). The values were much lower than those described for other blue cheese varieties [8,26]. The salt/moisture concentration is variable depending on the type of cheese, in most cases between 2–6%, which allows the growth of starter cultures and

limits the development of some pathogens. Similarly, low salt/moisture levels contribute to the hydration and solubilisation of caseins and the melting of curd granules [27]. After 2–3 months of ripening, blue cheeses have a salt/moisture content of around 6–8%, which is compatible with the development of *P. roqueforti*, while inhibiting the appearance of other contaminating moulds such as *Geotrichum candidum* [28]. In turn, these high salt/moisture levels inhibit the melting and compaction of the curd grains, maintaining a more open texture of the cheese mass and thus better development of blue mould. Cheeses salted in the partially drained curd (L5) had salt/moisture levels far from the optimal range for ripened blue cheeses.

The highest acidity values were described for dry-salted blue cheeses, which were very similar in all of them, although they differed significantly ($p < 0.05$) from those obtained for cheeses from L4 and L5 batches. The acidity of the latter samples was 38% and 26% lower than that of the dry-salted cheeses. These values could be explained by the lower salt/moisture concentration and higher humidity of the salted cheeses in brine and curd, which would allow a faster metabolism of lactic acid by the mould *P. roqueforti* [29–31]. In fact, the mass of the blue cheese salted in brine (L4) showed a greater and more homogeneous growth of *P. roqueforti* throughout the mass, while the dry-salted cheeses (L1, L2, and L3) showed less mould growth in the part closest to the rind. This was later confirmed by analysing the pH values of the cheeses, which showed that the cheeses salted in brine (L4) had higher pH values than the dry-salted and curd-salted cheeses. Lactic acid has the ability to inhibit several microbial populations; however, *P. roqueforti* is very tolerant [32].

The pH of blue cheeses increases during ripening due to the consumption of lactic acid by moulds and yeasts and the proteolytic process that takes place during ripening, releasing a large amount of alkaline compounds (amino acids and ammonium) that contribute to the increase in the pH of the cheese mass [33]. In other similar studies, it has been observed that the pH of cheeses increases with ripening, reaching values of 6.77 for Cabrales [34] or 6.84 for Gorgonzola [35]. In general, pH values were slightly lower than those described by these authors, although they were like those observed by others [8,36] who reported pH values of 5.97 and 5.80 for Valdeón and Danablu cheeses, respectively, for the same ripening time. The different salting methods used in this study do not seem to have a very significant effect on the pH values after one month of ripening; therefore, it is likely that the evolution of proteolysis was similar in the different batches of cheese. This could be related to the salt/moisture concentrations present in the cheeses, which are within the optimal range of action of the proteinases of the starter culture and other associated lactic acid bacteria, as well as the coagulation enzyme, native milk proteinases, and in particular, the proteinases and exo- and endopeptidases secreted by *P. roqueforti* [36,37].

Values of aw obtained in our study showed a behaviour very similar to that described for the salt/moisture relationship. The lowest values were described in the dry-salted cheeses (L1, L2, and L3), followed by those salted in brine (L4), while the highest values were obtained in the cheeses salted in the mass of curd (L5). Dry-salted cheeses showed significant differences ($p < 0.05$) with respect to the others, as well as differences between the other batches. These results were different from those described by other authors, who reported aw values for blue cheeses ranging from 0.880 to 0.925 [6,9–11]. However, in the studies reviewed, aw values corresponded to cheeses with a longer ripening period (2 to 4 months) and different degrees of salting. In a recent study on the Protected Geographical Indication of Valdeón blue cheese by Diezhandino et al. [8], they reported aw values of 0.945 for the cheeses after one month, although their salt/moisture ratio was much higher than that obtained in our study, confirming the influence of this last parameter on the final aw of the cheese. The extent and depth of proteolysis that blue cheeses undergo during ripening generate low molecular weight compounds, which also play a very important role in the immobilisation of free water [38,39]. As previously described, aw is a fundamental factor in controlling the growth of microbiota in cheese during ripening [40,41]. The values of aw obtained for the different batches are compatible with the growth of LAB and *P. roqueforti*, but also with some pathogenic bacteria or contaminating

moulds and yeasts that can be found in cheese. Dry-salted cheeses showed aw values of 0.970 but salted cheeses in brine (L4) and curd (L5) showed values of 0.982 and 0.992, respectively. These higher values would contribute to the implantation of contaminating moulds such as *G. candidum*, *Mucor* spp., or yeasts, which inhibit the germination and development of *P. roqueforti* while contributing to the cheese not acquiring the optimal sensory characteristics for its commercialization [42].

A correlation was observed between moisture content and luminosity, with the highest values of luminosity coinciding with blue cheeses with higher moisture content. The L4 and L5 cheeses showed similar moisture values, higher than those described for the dry-salted cheeses (L1, L2, and L3). In the latter, it was observed that the higher the salt concentration, the lower the moisture content and the luminosity [43]. In fact, Diezhandino et al. [44] reported that there is a negative correlation between luminosity and the total solids. Regarding the values obtained for the red/green coordinates (a*), significant differences were observed between the dry-salted cheeses (L1, L2, and L3) and the cheese salted in the mass of curd (L5) with respect to those salted in the brine (L4), with a greater tendency toward greenish tones in the dry-salted samples. These colours are associated with the presence of *P. roqueforti* in the cheese mass, whose evolution was quite similar in dry-salted and brine-salted cheeses. Colour parameters showed values slightly lower than those described by Diezhandino et al. [19] for Valdeón cheese with the same ripening time and by Kneifel et al. [45] for Roquefort cheese. The results for the yellow/green coordinates (b*) showed a predominance of yellow colour, which is related to the presence of carotenoids in the milk. In fact, the amount of carotenoids present in cheeses varies according to different factors, with one of the most important being the type of feed given to the cows [46]. In addition to feeding, the loss of moisture during the ripening process contributes to the increase in the yellow colour of the cheeses, with the b* values not being influenced by the type of salting applied to the cheeses, as described by Pavia et al. [47].

The results obtained for hardness, gumminess, and chewiness were lower than those described by Diezhandino et al. [19] for Valdeón cheese, while the fracturability results were slightly higher. A direct relationship has been reported between the moisture content and the hardness of the cheese because the decrease in water content favours a greater concentration of caseins in the matrix, which increases the hardness [48,49]. This fact would explain why the salted cheeses in brine (L4) and in the mass of curd (L5) with the highest moisture content were those with the lowest hardness. The same effect was observed by Diezhandino et al. [44] in Valdeón cheese ripened for two months, with a positive correlation between pH and hardness. On the other hand, the higher hardness values of cheeses salted in brine, compared with those salted in partly drained curd, would be determined by the differences in their salt content. The salt concentration in the salted cheeses in brine (L4) and dry-salted cheeses (L1, L2, and L3) led to an increase in the ionic strength of the cheese matrix with consequent dehydration of the proteins, resulting in a strengthening of the protein–protein interactions and a greater hardness of the cheese. This effect was greater in dry-salted cheeses, which had lower water content and a higher salt/moisture ratio [19,50]. On the contrary, in the cheeses salted in the mass of curd (L5), low salt concentrations would contribute to the fixation of water to the proteins, keeping the protein network more hydrated, with lower resistance to deformation and lower fracturability values [51]. Gumminess and chewiness values followed the same trend as hardness, as they are defined by it and are all directly related to moisture and protein content [52]. Cheeses with a lower salt content were characterised by the fact that less force was required during chewing and subsequent swallowing. Adhesiveness, cohesiveness, and springiness values were slightly lower than those described by Diezhandino et al. [19] for Valdeón cheese. These authors associate greater cheese adhesiveness with a higher degree of proteolysis, which contributes to an increase in the interactions of proteins with water and with other non-protein components, resulting in a more swollen and hydrated mass, as well as a high-fat content. Given that the blue cheeses in this study had similar pH values, it would be expected that the extent of proteolysis would develop in a similar way,

with possible differences between cheeses being compensated by their water content, thus contributing to all cheeses having similar adhesion values, regardless of the salting process. The lack of differences in cohesiveness and springiness between the cheese batches could have been influenced by the intense deformation conditions applied to the samples during the texture profile analysis. The springiness of cheese is related to the concentration of casein and its intra- and intermolecular bonds: the more numerous they are, the more difficult it is to deform the network and therefore the higher the springiness [53]. Therefore, proteolytic reactions on the casein network may also have contributed to the low springiness values.

Cheeses L1 and L2 showed a very similar development and uniform distribution of the mould in their mass, whereas dry-salted cheeses after 48 h (L3) and salted in brine (L4) showed less growth and homogeneity in the distribution of the blue vein. In both cheeses, although more pronounced in those dry-salted for 48 h, greater mould growth was observed in the central part of the cheese compared with the area closest to the rind, which would be related to a higher salt concentration in these areas, preventing the proper development of the mould [28,54]. The degree of moulding is related to the growth of *P. roqueforti* in the cavities of the cheese, while homogeneity refers to the even distribution of the mould throughout the mass [31]. The homogeneous distribution of salt in dry-salted or brine-salted cheeses takes time and is influenced by the water content of the curd. In the case of dry-salted (L3) and brine-salted cheeses (L4), salting was carried out after 48 h and 36 h, which would probably lead to a slowing down of the uptake and distribution of salt by these cheeses. The behaviour of the mould in the cheeses salted in the mass of curd (L5) is particularly noteworthy, as these were the cheeses with the lowest levels of mould growth and blue vein homogeneity. These cheeses, despite having a moisture content and optimum salt/moisture and aw values for mould development, were characterised by a more closed and compact mass, which prevented the correct development and distribution of the *Penicillium* in the few cavities present in blue cheeses.

Mould growth occurred in all the cheeses studied, which together with optimal environmental conditions, allowed the release and activity of fungal proteinases, peptidases, and lipases, contributing to the flavour and similar aromatic profile [31]. However, the degree of saltiness of the cheeses could play an important role in flavour evaluations. Thus, the dry-salted cheeses at 12 h (L1) and those salted in the mass of curd (L5) were also the first selected by the tasters. In general, blue cheeses are characterised by a high concentration of sodium chloride, which is necessary for the control of contaminating microorganisms and the correct development of *P. roqueforti* during ripening [55]. However, the hypertension problems of today's society have led many consumers to reduce salt in their diet. This may have contributed to the fact that a significant proportion of the tasters, accustomed to a low-salt diet, preferred blue cheeses with a lower salt content (L1 and L5). Blue cheeses with a higher salt/moisture ratio and salt content received lower ratings for flavour, probably due to a greater slowing down of the activity of the microbiota and the biochemical processes involved in the formation of sapid compounds [48]. Finally, the tasters also noted significant differences ($p < 0.05$) between the different cheeses in terms of hardness. The tasters preferred dry-salted cheeses for 12 h (L1) and 24 h (L2), as well as those salted in the partly drained curd (L5), to dry-salted cheeses for 48 h (L3) and in brine (L4). These results did not correlate with those previously described for hardness in the analysis of the instrumental texture profile of the cheeses. The overall impression was very similar in all the cheeses; consequently, although the type of salting applied to the cheese could have some influence on the sensory attributes evaluated individually, at a global level the consumers of the tasting panel were not able to detect these differences. This fact is of great importance for blue cheese producers since modifying the salting process under the conditions described in this work would not have a negative impact on the sensory characteristics of blue cheese.

## 5. Conclusions

The salting method used had a significant influence on the moisture and salt content of the blue cheeses at the end of the ripening process, with the cheeses salted in brine and the cheeses salted in curd having the highest values of moisture and the lowest values of salt/moisture, water activity, and titratable acidity compared with the dry-salted cheeses. On the other hand, the type of salting did not modify the CIELab parameters of the colour of the blue cheeses, except for the red/green coordinates (a*), which showed less tendency toward greenish colours in the salted curd cheeses. Cheeses salted in curd and brine had lower values for hardness, fracturability, and gumminess than those dry-salted, requiring less chewing to swallow. Tasters preferred the dry-salted cheeses after 12 and 24 h due to their better flavour and textural characteristics, as well as the more homogeneous distribution of mould in the cheese. However, from a sensory point of view, and regardless of the salting process used, no differences were found in the overall impression of the blue cheeses, which all scored around 7. The manual application of dry salt to cheeses is very labour-intensive and, at the same time, results in a large loss of salt. Therefore, and based on the results of this work, blue cheeses with similar characteristics could be obtained by salting in brine, with the need to adjust parameters such as the draining time of the curd before salting and the salting time. This would allow a better standardisation of the salting process, require less labour, and be more economical and environmentally friendly by reducing salt losses during the salting of the cheese.

**Author Contributions:** Conceptualization, N.L.G., J.M.F. and M.E.T.; methodology, D.A., P.C.-F., B.P., J.M.F. and M.E.T.; software, N.L.G., D.A. and B.P.; formal analysis, N.L.G., B.P., D.A. and P.C.-F.; investigation, N.L.G., D.A. and P.C.-F.; resources, N.L.G., J.M.F., B.P. and M.E.T.; writing—original draft preparation, N.L.G., J.M.F. and M.E.T.; writing—review and editing, N.L.G., J.M.F. and M.E.T.; and supervision, J.M.F. and M.E.T. All authors have read and agreed to the published version of the manuscript.

**Funding:** This research received no external funding.

**Institutional Review Board Statement:** Not applicable.

**Informed Consent Statement:** Not applicable.

**Data Availability Statement:** The data presented in this study are available on request from the corresponding author.

**Acknowledgments:** The authors would like to thank the Quesos La Peral, S.L. cheese factory (La Peral-Illas, Asturias, Spain) for their cooperation in supplying milk and in the production and ripening of the cheeses.

**Conflicts of Interest:** The authors declare no conflicts of interest.

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
