# Peer review of "Influence of Salting on Physicochemical and Sensory Parameters of Blue-Veined Cheeses"

_2624-862X, doi:10.3390/dairy5010008_

Round 1

Reviewer 1 Report

Comments and Suggestions for Authors

Lines 31 to 42: There is an overlap of meanings concerning curd and/or cheese, as well as repetitions. Is there a reason to discriminate curd from cheese, as it looks like there is, the authors may make one approach (salting curd, or salting cheese, or brine), explaining the effects of salt and avoid repetition if not necessary.

Author Response

Authors: Thank you so much for the comments and suggestions.

Reviewer 1: Lines 31 to 42: There is an overlap of meanings concerning curd and/or cheese, as well as repetitions. Is there a reason to discriminate curd from cheese, as it looks like there is, the authors may make one approach (salting curd, or salting cheese, or brine), explaining the effects of salt and avoid repetition if not necessary.

Authors: A distinction has been made in the text between types of salting with salting of partially drained curd before moulding (salting in the mass of curd) (for example, in lines 13, 17, 19, 33, 52-53, 88, 116, as well as each time it is referred to (highlighted in red throughout the text).

Reviewer 2 Report

Comments and Suggestions for Authors

Dear author,

The topic of this study is worthy but there are some comments on the manuscript and these comments should be taken into consideration before publishing.

General comments:

The manuscript was written well but there are two points, and some statements are needed to rephrase such as:

·         In the line 95 (3.84% fat, 3.84%) this ratio of fat was write twice

·         In the line 269,270,271 (The differences observed in the salt concentration between the cheeses salted for 48 h (L3) and those dry-salted for 12 h (L1) and 24 h (L2) could be related to the degree of draining of the curds before salting) I think this difference related to the open structure of the other surface that didn’t salted yet, the salt was absorbed from this surface that still opened for 48 h before salted.

Author Response

Reviewer 2:

Authors: Thank you so much for the comments and suggestions.

General comments:

The manuscript was written well but there are two points, and some statements are needed to rephrase such as:

  • In the line 95 (3.84% fat, 3.84%) this ratio of fat was write twice

Authors: The repetition has been corrected in line 98 of the new version. Thank you very much.

  • Reviewer 2: In the line 269,270,271 (The differences observed in the salt concentration between the cheeses salted for 48 h (L3) and those dry-salted for 12 h (L1) and 24 h (L2) could be related to the degree of draining of the curds before salting) I think this difference related to the open structure of the other surface that didn’t salted yet, the salt was absorbed from this surface that still opened for 48 h before salted.

Authors: The cheeses from batch L3 are not salted until 48 hours after removal from the mould and the structure is still effectively open and therefore porous. This has been taken into account in the lines of the new version (line 279 of the new version).

Reviewer 3 Report

Comments and Suggestions for Authors

There are some miner comments as follows:

  • Line 95; delete the repeat number of 3.84%.
  • Line 97: What is the concentration of calcium chloride used?
  • Line 165: How was the elasticity measured by the texture analyzer? I think it is springiness; it should also be checked in the results and discussion section.
  • Table 3: What are the units of springiness (not elasticity), cohesiveness, gumminess, and chewiness?

Author Response

Reviewer 3:

Authors: Thank you so much for the comments and suggestions.

Comments and Suggestions for Authors

There are some miner comments as follows:

Reviewer 3: Line 95; delete the repeat number of 3.84%.

Authors: The repetition has been corrected in line 98 of the new version. Thank you very much.

Reviewer 3: Line 97: What is the concentration of calcium chloride used?

Authors: 30% (w/v) calcium chloride solution. This has been included in line 100 of the new version.

Reviewer 3: Line 165: How was the elasticity measured by the texture analyzer? I think it is springiness; it should also be checked in the results and discussion section.

Authors: It is indeed springiness, thank you, it was a translation error. Thank you. This has been corrected (lines 169, 234, Table 3, 405, 413, 415, 418, 419.

Reviewer 3: Table 3: What are the units of springiness (not elasticity), cohesiveness, gumminess, and chewiness?

Authors: Springiness is a percentage; cohesiveness is dimensionless, gumminess (N), adhesiveness (N.s), and chewability (N). This was included in table 3 of the new version.

Reviewer 4 Report

Comments and Suggestions for Authors

The research presents an interesting subject, with a well-described methodology, coherent analyses, and the conclusion responds to the proposed objective.

Some more specific comments:

1. The references used throughout the article are mostly not very current, research with this type of food matrix is quite widespread, so the references can be updated.

2. The number of tasters for sensory analysis could be greater, research articles generally involve 100 tasters to mitigate evaluation errors.

3. How many times was the experiment carried out? An experiment was carried out on an industrial scale, but even so, at least one more experiment would be necessary to repeat the analyzes in the laboratory to reduce experimental errors/deviations.

4. Figure 1 has low resolution and small font, making the information difficult to read.

Comments on the Quality of English Language

English needs to be revised, there are expressions in the first person plural throughout the text and some expressions that need to be improved.

Author Response

Reviewer 4:

Authors: Thank you so much for the comments and suggestions.

Comments and Suggestions for Authors

The research presents an interesting subject, with a well-described methodology, coherent analyses, and the conclusion responds to the proposed objective.

Some more specific comments:

  1. Reviewer 4: The references used throughout the article are mostly not very current, research with this type of food matrix is quite widespread, so the references can be updated.

Authors: several recent references have been included in the new version and are highlighted in red in the text and in the reference list.

  1. Reviewer 4: The number of tasters for sensory analysis could be greater, research articles generally involve 100 tasters to mitigate evaluation errors.

Authors: The tasting was led by university students and professors, some of whom had some experience of sensory analysis of cheese, so the error is likely to be lower than expected simply because of the number of tasters.

  1. Reviewer 4: How many times was the experiment carried out? An experiment was carried out on an industrial scale, but even so, at least one more experiment would be necessary to repeat the analyzes in the laboratory to reduce experimental errors/deviations.

Authors: 5 batches of cheeses were produced in duplicate (10 cheeses for sampling). Chemical analyses were performed in triplicate, and 9 replicates for texture analysis and 12 replicates for colour determination were performed on each of the 10 cheeses, so the authors consider that the number of replicates was not low.

  1. Reviewer 4: Figure 1 has low resolution and small font, making the information difficult to read.

Authors: The resolution of figure 1 is not that it is not good, it is that it loses resolution when converted to PDF. So it could be sent to the editor in a different format if you think it is appropriate.

Comments on the Quality of English Language

Reviewer 4: English needs to be revised, there are expressions in the first-person plural throughout the text and some expressions that need to be improved.

Authors: The English language has been revised throughout the text to correct any incorrect expressions. The changes have been highlighted in red throughout the text.

Round 2

Reviewer 4 Report

Comments and Suggestions for Authors

The questions have all been answered and I consider it suitable for publication.